# The Microsurgical Resection of an Arteriovenous Malformation in a Patient with Thrombophilia: A Case Report and Literature Review

**DOI:** 10.3390/diagnostics14232613

**Published:** 2024-11-21

**Authors:** Corneliu Toader, Felix-Mircea Brehar, Mugurel Petrinel Radoi, Matei Serban, Razvan-Adrian Covache-Busuioc, Luca-Andrei Glavan, Alexandru Vlad Ciurea, Nicolaie Dobrin

**Affiliations:** 1Department of Neurosurgery, “Carol Davila” University of Medicine and Pharmacy, 020021 Bucharest, Romania; corneliu.toader@umfcd.ro (C.T.); matei.serban2021@stud.umfcd.ro (M.S.); razvan-adrian.covache-busuioc0720@stud.umfcd.ro (R.-A.C.-B.); luca-andrei.glavan0720@stud.umfcd.ro (L.-A.G.); prof.avciurea@gmail.com (A.V.C.); 2Department of Vascular Neurosurgery, National Institute of Neurology and Neurovascular Diseases, 077160 Bucharest, Romania; 3Department of Neurosurgery, Clinical Emergency Hospital “Bagdasar-Arseni”, 041915 Bucharest, Romania; 4Department of Neurosurgery, Sanador Clinical Hospital, 010991 Bucharest, Romania; 5“Nicolae Oblu” Clinical Hospital, 700309 Iasi, Romania; dobrin_nicolaie@yahoo.com

**Keywords:** thrombophilia, arteriovenous malformation, neurosurgery, Prothrombin G20210A mutation, microsurgical resection

## Abstract

**Background/Objectives:** Arteriovenous malformations (AVMs) are complex vascular anomalies that can present with significant complications, including intracranial hemorrhage. This report presents the case of a 36-year-old female with Prothrombin G20210A mutation-associated thrombophilia, highlighting its potential impact on AVM pathophysiology and management. **Methods:** The patient presented with a right paramedian intraparenchymal frontal hematoma, left hemiparesis, and seizures. Cerebral angiography identified a ruptured right parasagittal frontal AVM classified as Spetzler–Martin Grade II. A right interhemispheric frontal craniotomy was performed, enabling microsurgical resection of the AVM. Intraoperative findings included evacuation of a subcortical hematoma and excision of a 20 mm AVM nidus with arterial feeders from the A4 segment of the anterior cerebral artery and a single venous drainage into the superior sagittal sinus. **Results:** Postoperative recovery was favorable, with significant neurological improvement. The patient demonstrated resolution of hemiparesis and a marked reduction in seizure activity. The hypercoagulable state associated with Prothrombin G20210A mutation was identified as a contributing factor in the thrombosis of the AVM’s draining vein, potentially leading to increased venous pressure, rupture, and hemorrhage. **Conclusions:** This case underscores the importance of recognizing thrombophilia in patients with AVMs for optimal surgical planning and complication management. Despite the challenges posed by the hypercoagulable condition, microsurgical resection proved to be a viable and effective treatment option. Further research is warranted to elucidate the relationship between thrombophilic disorders and AVMs to enhance patient management strategies and outcomes.

## 1. Introduction

Vascular anomalies encompass a wide range of conditions, from benign birthmarks to potentially life-threatening abnormalities. These anomalies are generally classified into two main categories: vascular malformations and proliferative vascular lesions (tumors) [1]. McCormick further refined this classification of vascular malformations into four distinct groups: venous malformations, arteriovenous malformations (AVMs), telangiectases, and cavernous malformations [2]. Cerebral AVMs are particularly notable as they involve abnormal connections between arteries and veins, bypassing the capillary system entirely [3]. Among the various types of intracranial vascular malformations, AVMs carry a significantly higher risk of hemorrhage [4].

Although the exact cause of AVMs remains unclear, they are widely believed to be multifactorial and congenital in origin. Their formation may involve the dysregulation of vascular endothelial growth factor receptor-mediated endothelial proliferation, coupled with cytokine-driven vessel remodeling processes [5]. Brain AVMs are rare, with an estimated prevalence of 0.05% in the general population, and they exhibit no significant gender preference. They are typically diagnosed in young adults, with the average age being 31 years or younger [6]. Berman et al. conducted a meta-analysis suggesting that due to variations in the detection rates of asymptomatic AVMs, the most accurate prevalence estimates are derived from symptomatic cases, with an average detection rate of 0.94 per 100,000 person-years [7]. The most common site for AVMs is the supratentorial region, though they can occur in various areas of the brain, including the frontal lobes (24%), temporal lobes (19%), parieto-occipital lobes (19%), cerebellum (20%), deep structures (8%), periventricular region (7%), and brainstem (3%) [8].

The clinical presentation of brain AVMs varies significantly, ranging from mild headaches to more severe manifestations, such as seizures, muscle weakness, visual disturbances, and cognitive impairment. Hemorrhage is often the initial symptom, with approximately 50% of cases presenting with bleeding. A hemorrhage from an AVM can lead to stroke and other serious neurological complications, necessitating urgent medical intervention [4].

Treatment strategies for AVMs typically involve a combination of endovascular embolization, radiosurgery, and microsurgical resection, with the choice of intervention often guided by the Spetzler–Martin grading system [9] or the Lawton–Young (Supplementary Spetzler–Martin) grading system [10]. The primary goal of treatment is to prevent spontaneous intracranial hemorrhage [11]. Grade I AVMs carry minimal surgical risk, while Grade II AVMs, with a low surgical risk score of two, are also considered suitable for surgical intervention. In contrast, Grade III AVMs present moderate surgical risks, and Grades IV and V, which score three and five points, respectively, are generally deemed inoperable. Importantly, the microsurgical resection of unruptured Spetzler–Martin Grade I and II AVMs has been shown to reduce long-term morbidity and enhance quality-adjusted life years (QALYs) compared to conservative management [12]. Despite the superior predictive accuracy of the Supplementary Spetzler–Martin system for early and late postoperative outcomes, the original Spetzler–Martin grading system remains widely utilized due to its familiarity and popularity [13].

## 2. Case Presentation

A 36-year-old female patient with a known history of thrombophilia (FII G20210A mutation) was transferred to our clinic for a right paramedian intraparenchymal frontal hematoma. Our patient presented with left hemiparesis and comitial crisis. A cerebral angiography was performed, which showcased a ruptured right parasagittal frontal AVM, Martin–Spetzler II (Figure 1, Figure 2 and Figure 3). In the 2D digital subtraction angiography (DSA), the parasagittal frontal AVM was clearly visualized, with abnormal vascular connections highlighted. The images captured both the arterial and venous phases. During the arterial phase, the feeding arteries supplying the AVM were distinctly visible, while the venous phase showed the early venous drainage of the malformation. The dense vascular network, typical of AVMs, was prominently observed, with abnormal connections between the arteries and veins. These imaging details were critical for assessing the AVM’s structure, including the arterial feeders and venous outflow, which played an essential role in guiding the surgical and interventional treatment planning.

This figure presents a combination of 3D reconstructed rotational angiographic images and digital subtraction angiography (DSA) scans, illustrating a brain AVM from various perspectives. The anterior and posterior views highlight the complex vascular network, showing the feeding arteries and the AVM nidus located in the parasagittal region. The intricate connections between the arteries and veins are visible, emphasizing the abnormality characteristic of AVMs. The central section contains additional reconstructions and graphical data, providing volumetric information about the AVM and surrounding vasculature, enhancing the spatial understanding necessary for precise treatment planning. The lateral view on the far right emphasizes the abnormal vascular connections and provides a detailed depiction of the AVM’s relationship with the surrounding brain vessels, essential for understanding its orientation and depth. These comprehensive images offer critical insights into the anatomical and hemodynamic features of the AVM, aiding in the planning of treatment options such as microsurgical resection or embolization. The detailed visualization of the arterial feeders, venous drainage, and nidus structure is vital for assessing rupture risk and guiding clinical decisions.

A right interhemispheric frontal craniotomy was performed. The dura mater was circumferentially suspended and incised with a medial pedicle. Upon entering the precentral gyrus, a subcortical hematoma, approximately 1 cm deep and consisting of around 20 mL of clotted and liquid blood, was encountered. The hematoma was evacuated both spontaneously and through saline lavage. Following the evacuation, the brain resumed its normal pulsations.

A corticectomy was then performed in the posterior third of the F1 gyrus, proceeding medio-inferiorly to the level of the free edge of the falx cerebri. Two feeders from the A4 segment of the anterior cerebral artery were identified at this point; they were coagulated and sectioned. The nidus, measuring approximately 20 mm, was located posterior to the feeders and anterior to the hematoma. The nidus was excised, and a single venous drainage into the superior sagittal sinus was intercepted and coagulated. 

A slight cerebral collapse was noted. Hemostasis was achieved using electrocoagulation, Surgicel, and tamponade. The bone flap was then replaced over an epidural drain, which was externalized through a burr hole. The wound was closed in anatomical layers, and a dressing was applied.

The patient experienced a favorable neurological recovery postoperatively under neurorehabilitation therapy (Figure 4). 

Our patient continued to come to routine follow-up evaluations (Figure 5).

This figure shows an 8-month follow-up CT scan of the patient after brain AVM treatment. The axial, coronal, and sagittal views reveal normal brain structures, with no evidence of hemorrhage or complications such as hydrocephalus or mass effect. The images confirm a stable postoperative outcome, indicating a favorable recovery without signs of recurrence or new abnormalities.

## 3. Discussion

A notable feature of this case is the presence of thrombophilia caused by the Prothrombin G20210A mutation, also known as the factor II mutation, which is among the most prevalent genetic causes of thrombosis, affecting 0.7–4% of the general population [14,15]. This mutation is characterized by a single nucleotide substitution at position 20210 of the prothrombin gene (F2) on chromosome 11, where guanine (G) is replaced by adenine (A) [16]. Carriers of the Prothrombin G20210A mutation, particularly those with a history of venous thrombosis, tend to exhibit increased thrombin generation, independent of plasma factor II levels, thereby significantly raising the risk of thrombus formation. Homozygous carriers face a 20-fold higher risk of thrombotic events, while heterozygous carriers have up to a 2.5-fold increased risk of venous thromboembolism (VTE) compared to noncarriers [17].

Patients with AVMs complicated by thrombosis may present with symptoms that mimic subacute or occult hemorrhage, including seizures, headaches, or neurological deficits [18]. A prevailing hypothesis for spontaneous thrombosis in AVMs is that the retrograde thrombosis of the single draining vein results in occlusion. Other factors, such as diminished blood flow or a hypercoagulable state, may also contribute to spontaneous thrombosis, which has been reported in 84% of such cases [19].

The potential role of thrombophilia in AVM rupture has been well documented, but recent advancements in the understanding of the underlying biological mechanisms warrant further discussion. Hypercoagulable states, such as those induced by the Prothrombin G20210A mutation, are known to increase the risk of venous thrombosis, which can precipitate AVM rupture by causing venous hypertension and reduced cerebral venous drainage. Recent studies, including work by Chen et al. (2023) [1], have identified increased venous pressure and thrombosis in the draining veins of AVMs as key contributors to nidus rupture. The resulting perinidal edema and venous congestion may further exacerbate these risks. Expanding our understanding of these hemodynamic changes provides valuable insight into how thrombophilia modulates AVM behavior and rupture risk. By including these details, we emphasize the need for personalized treatment approaches in patients with thrombophilic conditions undergoing AVM resection.

Thrombophilia in female patients with brain AVMs adds a significant layer of complexity to clinical management due to the increased risk of both thrombosis and hemorrhage. The studies summarized in Table 1 demonstrate that thrombophilic conditions, such as the Prothrombin G20210A mutation, markedly elevate the risk of venous thrombosis, which can lead to severe complications, including AVM rupture and intracranial hemorrhage. In female patients, particularly those of reproductive age, thrombophilia may exacerbate the hypercoagulable state, further complicating the natural progression of AVMs. This predisposition heightens the likelihood of seizures, neurological deficits, and hemorrhagic events.

The anatomical characteristics of AVMs—such as the nidus size, the presence of single or multiple arterial feeders, and venous drainage patterns—are critical factors in assessing the risks posed by thrombophilia in these cases. Treatment strategies, including microsurgical resection, transvenous embolization, and Onyx-based embolization, must be carefully tailored to the patient’s thrombophilic profile and the specific AVM’s location and complexity. For instance, embolization techniques may be optimized to reduce the risk of venous thrombosis, which is more prevalent in patients with thrombophilic disorders.

While the association between thrombophilia and AVMs is known, recent studies have shed light on the complexity of this relationship. Thrombophilic disorders such as Factor V Leiden, Protein C and S deficiencies, and the Prothrombin G20210A mutation all contribute to an elevated risk of venous thrombosis, which can alter the natural history of AVMs by increasing venous pressure and contributing to nidus rupture. For example, Agyemang et al. (2022) [2] demonstrated that hypercoagulability in AVM patients may lead to the spontaneous thrombosis of the draining veins, causing venous stasis and rupture. The interplay between these genetic mutations and AVM pathophysiology remains an area ripe for further exploration, particularly in how different thrombophilic conditions may necessitate tailored treatment strategies. This case underscores the importance of multidisciplinary care in managing AVMs, particularly in patients with underlying thrombophilic disorders.

Given the heightened risk of both hemorrhage and thrombotic events, the management of female patients with AVMs and thrombophilia requires a multidisciplinary approach. Neurosurgeons, neurointerventionalists, and hematologists must collaborate to balance the risks of anticoagulation with the need for effective AVM treatment. Overall, these studies emphasize the importance of personalized, case-specific treatment plans to optimize outcomes and reduce the risks of life-threatening complications in this patient population.

Table 1 presents a synthesis of key studies examining thrombophilia’s impact on brain AVMs, covering variables such as patient characteristics, the AVM location, the rupture status, and treatment methods. The findings showcase the range of complications associated with thrombophilic states in AVM cases, emphasizing the significance of individualized management strategies. These data highlight the necessity of tailoring treatment approaches to the specific risk profiles of patients, especially when hypercoagulability may increase surgical risks, underscoring the importance of a personalized approach in managing these complex cases.

Thrombosis in the draining veins of an AVM can result in stagnation and elevated venous pressure, potentially causing the fragile vessels within the AVM to rupture and lead to hemorrhage [9]. Furthermore, a perinidal edema could develop due to such elevated pressure, causing bleeding into nearby brain tissue. Conditions that promote thrombosis, such as hypercoagulable states (e.g., COVID-19 or genetic thrombophilias), further increase this risk by blocking drainage, leading to obstruction and significantly increasing the risk of hemorrhage [10,11]. We believe that, in the case of our patient, the thrombosis of her single venous drainage into the superior sagittal sinus occurred, which led to the development of the hemorrhage. AVM drainage veins often narrow and eventually thrombose due to local endothelial damage induced by turbulent blood flow and mechanical shearing stress exerted upon endothelial surfaces [12,13,14]. Others suggest that narrowing may also be caused by an increased shunt volume or elevated venous pressure [15]. Although this report focuses on the Prothrombin G20210A mutation, it is essential to consider how this mutation compares with other thrombophilic conditions. For instance, Factor V Leiden, another common thrombophilic mutation, is associated with a higher risk of venous thromboembolism and potentially similar complications in AVM patients. Studies by Sundquist et al. (2015) [16] and Favaloro et al. (2019) [17] have explored the differential risks associated with various genetic thrombophilias, suggesting that the Prothrombin G20210A mutation may present a slightly lower risk profile compared to Factor V Leiden, but still significantly increases the likelihood of venous thrombosis. This broader comparison allows us to place the Prothrombin G20210A mutation within the larger context of thrombophilic disorders, enabling clinicians to better assess the risks and plan individualized treatment protocols for AVM patients with diverse genetic backgrounds.

Moreover, the FII G20210A mutation is known to increase the risk of developing deep vein thrombosis (DVT) and pulmonary embolism [18]. Microsurgical resection is widely considered one of the most successful interventions for AVMs, boasting high rates of successful obliteration while simultaneously protecting neurological function [19]. Although the thrombophilia contributed to the difficulty of the surgical procedure, we opted for a microsurgical approach for this patient, which resulted in very positive outcomes, with the patient’s neurological status improving significantly after the procedure.

Recent reviews, such as those by Winkler et al. (2018) [7], suggest that microsurgical AVM resection carries significant risk, even in cases with favorable anatomy. It is important to note that while this particular case resulted in a successful outcome, similar interventions may not always achieve the same results, particularly when thrombophilia increases the risk of intraoperative and postoperative thrombotic complications. By addressing these broader risks, we aim to provide a more balanced perspective on the challenges of managing AVMs in thrombophilic patients, which extends beyond this individual case.

To provide a broader perspective on potential outcomes, we have considered surgical risks associated with AVM resection in patients with thrombophilia, referencing similar cases in the literature. Studies indicate that these patients may face an increased risk of perioperative complications, such as venous thrombosis and postoperative hemorrhage, due to their hypercoagulable state [20]. While our patient experienced a favorable outcome, it is essential to acknowledge that outcomes in similar cases may vary depending on individual factors such as the AVM’s location, grade, and complexity, as well as the patient’s overall thrombophilic profile.

The absence of a control group in this case report naturally limits our ability to definitively isolate the specific impact of the Prothrombin G20210A mutation on the management of AVMs. However, our primary objective was to present the clinical trajectory and postoperative outcomes of this rare and complex case, rather than to establish causality. Recent studies, such as those by Lin et al. (2019) [6] and Chen et al. (2023) [3], highlight that hypercoagulable states like thrombophilia can amplify the risk of AVM rupture through mechanisms like venous outflow obstruction and elevated venous pressure. While these insights are valuable, further research, including larger cohort or case–control studies, is essential to precisely determine the role of genetic thrombophilias such as Prothrombin G20210A in AVM management. This case contributes to the growing body of evidence and underscores the critical need for heightened vigilance in identifying hypercoagulability in AVM patients to optimize treatment strategies.

Recent studies have advanced our understanding of how thrombophilia contributes to AVM pathophysiology, particularly by exacerbating venous hypertension and elevating rupture risk. Hypercoagulable states, such as those associated with the Prothrombin G20210A mutation, have been implicated in a cascade of vascular changes that destabilize the delicate balance within AVM structures [1]. Research has demonstrated that increased venous pressure in thrombophilic patients often leads to retrograde venous thrombosis, causing drainage obstruction, perinidal edema, and heightened venous congestion. These hemodynamic changes create an environment that significantly raises the likelihood of AVM rupture, especially in patients with underlying thrombophilic conditions [22]. Additionally, recent work has examined the correlation between hypercoagulability and venous stasis within AVMs, showing that patients with genetic thrombophilias experience a higher incidence of spontaneous thrombosis in the primary draining vein, leading to local venous hypertension and structural instability within the nidus, thereby amplifying rupture risk [23].

Further studies have highlighted the role of hemodynamic stress in AVM rupture, noting how thrombophilic conditions intensify mechanical shearing forces on the endothelial lining of draining veins. This stress can lead to micro-tears and subsequent hemorrhagic events within AVMs. These findings suggest that thrombophilia, by raising the venous pressure and promoting venous stasis, significantly alters the natural progression and rupture risk of AVMs [24,25]. Expanding on these recent studies, our case underscores the complex interplay between hypercoagulable states and AVM management, highlighting the need for a multidisciplinary approach that integrates hematological assessment with neurosurgical planning. This approach is crucial for anticipating and mitigating the unique risks posed by thrombophilia, given its potential to increase venous congestion, impair drainage, and elevate rupture susceptibility in cerebral AVMs.

We acknowledge the limitation of this case report in its ability to generalize findings to a broader population, as it represents a single patient with a unique clinical profile. Although this report provides valuable insights into the management of AVMs in patients with thrombophilia, further studies involving larger patient cohorts are necessary to establish more robust clinical guidelines and to understand how thrombophilia influences AVM pathophysiology and surgical outcomes more broadly.

## 4. Conclusions

This case underscores the significant impact that thrombophilia, specifically the Prothrombin G20210A mutation, can have on the pathophysiology and management of cerebral AVMs. The hypercoagulable state associated with this genetic mutation may contribute to thrombosis within the draining veins of an AVM, leading to increased venous pressure, stagnation, and a heightened risk of hemorrhage. Recognizing the presence of thrombophilia in patients with AVMs is crucial for tailoring surgical strategies and anticipating potential complications.

Despite the challenges posed by the patient’s thrombophilic condition, microsurgical resection proved to be a successful intervention, resulting in significant neurological improvement and favorable postoperative outcomes. This suggests that, with careful planning and the consideration of the hypercoagulable state, microsurgical approaches remain viable for patients with AVMs complicated by thrombophilia.

Further research is warranted to deepen the understanding of the interplay between thrombophilic disorders and cerebral AVMs. Such studies could inform the development of optimized management protocols and surgical techniques that address the unique risks associated with hypercoagulable states, ultimately improving patient outcomes in this complex clinical scenario.

## Figures and Tables

**Figure 1 diagnostics-14-02613-f001:**
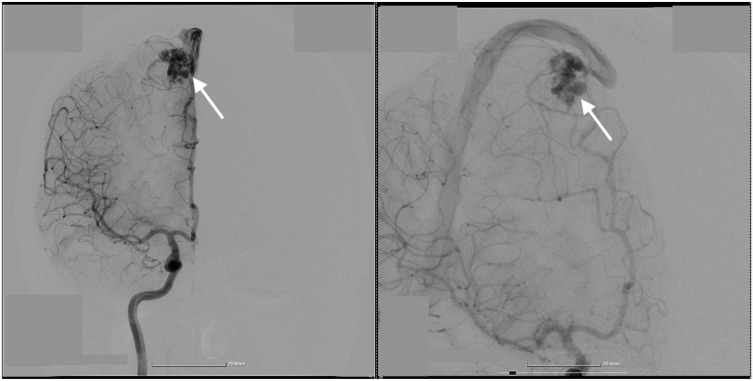
Preoperative 2D digital subtraction angiography. 2D DSA showcases the parasagittal frontal AVM, marked by the white arrows. The AVM, indicated by white arrows in both images, is lo-cated in the parasagittal region. The left image highlights the arterial phase, where the feeding arteries are visible as they supply the AVM. The right image captures the venous phase, demon-strating the early venous drainage of the AVM. The dense vascular network and abnormal con-nections between the arteries and veins, typical of an AVM, are clearly seen. These images provide critical details for assessing the AVM’s structure, including the arterial feeders and venous out-flow, which are essential for planning surgical or interventional treatment.

**Figure 2 diagnostics-14-02613-f002:**
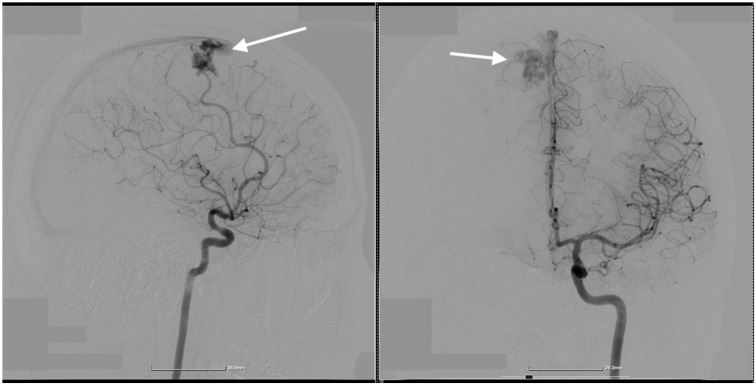
Preoperative 2D digital subtraction angiography in both the profile and frontal view which depicts the AVM, marked by the white arrows.

**Figure 3 diagnostics-14-02613-f003:**
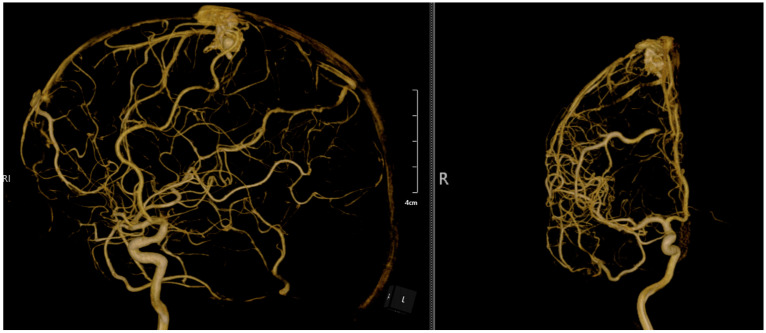
Preoperative 3D DSA rotational angiography, with the reconstruction of rotational DSA depicting the 3D topography of the arteriovenous malformation with an arterial A4 feeder from the anterior cerebral artery and single venous drainage into the superior sagittal sinus.

**Figure 4 diagnostics-14-02613-f004:**
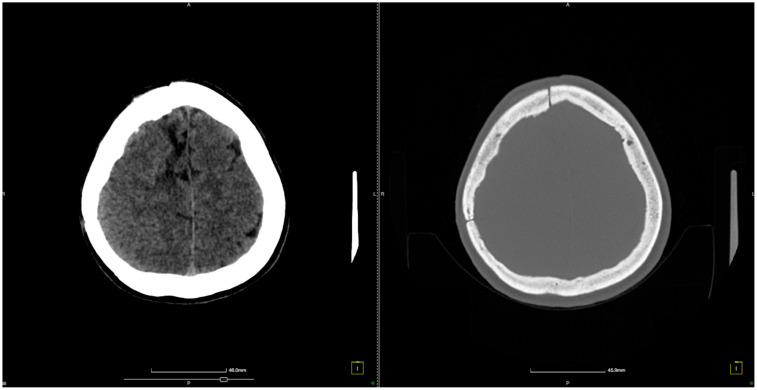
CT scan conducted just before patient discharge shows no signs of hemorrhage.

**Figure 5 diagnostics-14-02613-f005:**
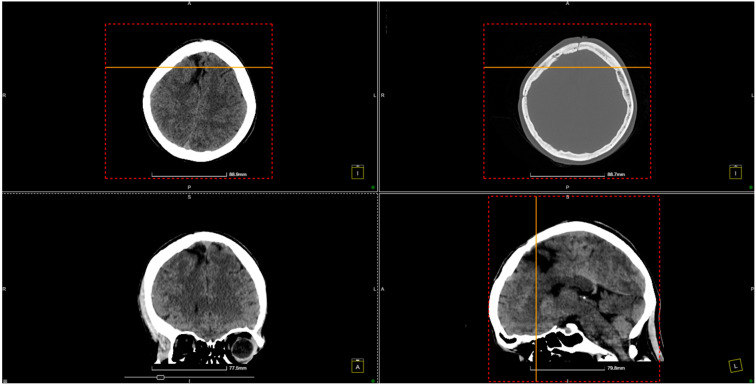
Eight-month follow-up CT scan indicates no hemorrhage.

**Table 1 diagnostics-14-02613-t001:** Summary of studies on thrombophilia in brain AVM patients.

Study	Number of Patients	Sex Ratio	AVM Localization	Rupture Status	AVM Diameter	Associated Complications	Arterial Feeder(s)	Venous Outflow(s)	Size of Nidus (cm)	Surgical Approach	Treatment Method	Characteristics of Thrombophilia
Chen et al. (2023) [20]	4	Mixed	Deep brain regions	All ruptured	0.79–2.56 cm	Hemorrhage, neurological deficits	Tiny arterial branches	Single draining vein	0.79–2.56	Transvenous embolization	Onyx embolization	Thrombophilia not detailed, but hemorrhage risks increased in hypercoagulable states
Massoud et al. (2022) [1]	4	Mixed	Supratentorial	Ruptured	0.79–2.56 cm	Hemorrhage, embolization-induced rupture	Small feeders with tortuous paths	Single draining vein	0.79–2.56	Transvenous embolization	Onyx-based TVE	TVE management helped control complications associated with thrombosis risks
Chen et al. (2021) [4]	5	Mixed	Deep brain	Ruptured	Variable	Neurological deficits, hemorrhage	Variable deep feeders	Single draining vein	Variable	Transvenous embolization	Combined approach (arterial and venous)	No direct thrombophilia analysis, but TVE showed good outcomes in hemorrhage control
Rodrigues et al. (2020) [5]	1	Male	Parieto-temporal	Thrombosed (spontaneous)	Not specified	Seizures, focal neurological deficits	Anterior cerebral artery	Parieto-temporal cortical vein	Not specified	Conservative management	Anticoagulation, dexamethasone	Thrombosis linked to hypercoagulable state induced by COVID-19, leading to spontaneous AVM thrombosis
Lin et al. (2019) [21]	45	Mixed	Supratentorial	36% ruptured	1.8–6.2 cm	Hemorrhage, seizures	MCA feeders	Multiple venous drainers	3.2	Superselective catheterization	Endovascular embolization	Thrombophilia not directly associated with but implicated in hemorrhage risk
Burkhardt et al. (2018) [7]	24	Mixed	Supratentorial	Mixed	Variable	Venous drainage issues	Small arterial feeders	Single draining vein	Variable	Transvenous embolization	Combined transarterial and transvenous embolization	No thrombophilia data, but complications noted with venous thrombosis
Rooij et al. (2011) [8]	24	Mixed	Frontal, occipital, parietal, temporal	Mixed	1–3 cm	Hemorrhage, seizures	MCA and ACA feeders	Single superficial draining vein	1–3	Curative embolization	Onyx embolization	Thrombophilia not directly analyzed, but embolization was complicated by venous reflux

## Data Availability

The data presented in this study are available on request from the corresponding author.

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
