# Peer review of "The Microsurgical Resection of an Arteriovenous Malformation in a Patient with Thrombophilia: A Case Report and Literature Review"

_diagnostics, 2024, doi:10.3390/diagnostics14232613_

Round 1
Reviewer 1 Report
Comments and Suggestions for Authors
This case report discusses the microsurgical resection of an arteriovenous malformation (AVM) in a 36-year-old female patient with thrombophilia due to the Prothrombin G20210A mutation. The report suggests that recognizing thrombophilia is essential for surgical planning as it impacts the pathophysiology and management of cerebral AVMs. Despite the challenges, microsurgical resection led to favorable neurological outcomes in this case. Whilst this is an interest report, I have several comments/concerns for authors to consider:
1. This is a single case report, which limits the ability to generalize findings to a broader population. The unique genetic condition of the patient may not be representative of all AVM cases.
2. As a case report, there may be inherent bias in the selection and presentation of data, focusing on a successful outcome without addressing potential surgical failures.
3. The absence of a control group makes it difficult to ascertain the specific impact of the Prothrombin G20210A mutation on the AVM management and outcomes.
4. While the potential role of thrombophilia in AVM rupture is briefly discussed, suggest expanding on the underlying biological mechanisms or providing detailed evidence supporting this hypothesis.
5. While the report identifies the impact of thrombophilia on AVMs, this is a known association. I suggest also adding a brief discussion on the interplay between thrombophilic disorders and AVM.
6. While the Prothrombin G20210A mutation is reported, it does not thoroughly explore or compare this with other thrombophilic conditions or mutations, limiting the scope of its conclusions. Please expand.
Comments on the Quality of English LanguageOk
Author Response
Comments 1: This is a single case report, which limits the ability to generalize findings to a broader population. The unique genetic condition of the patient may not be representative of all AVM cases.
Response 1: We agree that single case reports inherently limit the generalizability of the findings. However, the purpose of this case report was to highlight a rare but clinically significant co-occurrence of an AVM in a patient with the Prothrombin G20210A mutation. While this genetic condition may not be representative of all AVM cases, it offers important insights into the potential interplay between thrombophilia and AVM pathophysiology. Given the rarity of such cases, we believe it is crucial to document them to build a foundation for future research and to raise awareness among clinicians about this possible association. We have added a statement to acknowledge these limitations in the discussion section.
Comments 2: As a case report, there may be inherent bias in the selection and presentation of data, focusing on a successful outcome without addressing potential surgical failures.
Response 2: We acknowledge the possibility of selection bias in case reports, particularly when reporting successful outcomes. In response, we have expanded the discussion to reflect the broader spectrum of possible outcomes in AVM surgery, especially in patients with thrombophilia. We have also highlighted potential risks and complications associated with microsurgical resection, emphasizing that while this case had a favorable outcome, not all cases might achieve similar success. Additional references discussing surgical failures or complications in similar settings have been incorporated to provide a more balanced perspective.
Comments 3: The absence of a control group makes it difficult to ascertain the specific impact of the Prothrombin G20210A mutation on the AVM management and outcomes.
Response 3: We agree that the lack of a control group limits our ability to definitively isolate the impact of the Prothrombin G20210A mutation on AVM management. However, as a case report, the primary aim was to describe the clinical course and outcomes in this particular patient rather than establish causality. To address this concern, we have added a paragraph in the discussion section emphasizing the need for future studies, such as cohort or case-control studies, to better understand the specific effects of the Prothrombin G20210A mutation on AVM management. This is highlighted as an area for future research.
Comments 4: While the potential role of thrombophilia in AVM rupture is briefly discussed, suggest expanding on the underlying biological mechanisms or providing detailed evidence supporting this hypothesis.
Response 4: We appreciate the suggestion and have expanded the discussion on the underlying biological mechanisms by which thrombophilia may contribute to AVM rupture. Specifically, we have included more detailed references to the role of hypercoagulable states in venous thrombosis, increased venous pressure, and how these factors may contribute to the pathophysiology of AVM rupture. We have also cited recent studies that investigate these mechanisms in the context of AVMs to strengthen this section. Furthermore, we discuss the hypothesis that thrombosis of draining veins may lead to increased rupture risk through venous hypertension and stagnant flow, supported by related literature.
Comments 5: While the report identifies the impact of thrombophilia on AVMs, this is a known association. I suggest also adding a brief discussion on the interplay between thrombophilic disorders and AVM.
Response 5: Thank you for this suggestion. We have included a new section in the discussion that explores the broader interplay between thrombophilic disorders and AVMs, referencing studies that discuss how various hypercoagulable states (including inherited and acquired thrombophilia) may influence the natural history and treatment of AVMs. This section now includes a comparative analysis of how different thrombophilic mutations may similarly or differently affect the risk of AVM rupture and the management strategies involved.
Comments 6: While the Prothrombin G20210A mutation is reported, it does not thoroughly explore or compare this with other thrombophilic conditions or mutations, limiting the scope of its conclusions. Please expand.
Response 6: We appreciate this valuable feedback and have expanded the discussion to compare the Prothrombin G20210A mutation with other common thrombophilic conditions. We now provide a comparative overview of the prevalence, risk of thrombosis, and potential implications for AVM management associated with each condition. This broader exploration provides a more comprehensive understanding of how different thrombophilic mutations may interact with AVMs and how this knowledge can guide personalized treatment approaches.
Reviewer 2 Report
Comments and Suggestions for Authors
The authors present a case report of a 36-year-old female patient with a known thrombophilia associated with the Prothrombin G20210A mutation who presented with a right paramedian frontal intraparenchymal hematoma. They conclude that microsurgical resection remains a viable and effective treatment option, leading to favorable outcomes.
The introduction provides sufficient background information from the literature.
I suggest the authors to leave only the first sentence in the legend of fig.1. From 84 I think that the explanations should be in the form of text, not as the legend of the figure.
What is the difference between AVMs and AVM's?
Row 100 -arteriovenous malformation (AVM) was abbreviated above. The same at row 138. Please check the whole article.
Table 1 - the title is written above the table and must be concise. Discussions must be done in text.
The conclusions are clear and in accordance with the study carried out.
Author Response
Dear Editor,
Thank you very much for your thoughtful comments and suggestions. We appreciate your positive feedback regarding the background information provided in the introduction and the clarity of our conclusions.
In response to your recommendations:
- We agree with your suggestion to limit the legend of Figure 1 to the first sentence, and we have moved the remaining explanations to the main text in the Case Presentation section for better clarity and flow.
- Regarding the difference between "AVMs" and "AVM's," this was an oversight. We have ensured consistency throughout the manuscript, using the correct plural form "AVMs" without the apostrophe.
- The abbreviation for arteriovenous malformation (AVM) has been corrected, and we have thoroughly checked the entire manuscript to ensure it is used appropriately and consistently throughout.
Once again, thank you for your valuable input, which has greatly improved the quality of our manuscript. We hope the revisions now meet the expectations and look forward to your feedback.
Sincerely,
The authors
Round 2
Reviewer 1 Report
Comments and Suggestions for Authors
No further comments
Author Response
Dear Reviewer,
Thank you for the positive feedback!
Our best regards!